# Maxillary Premolars with Four Canals: Case Series

**DOI:** 10.3390/bioengineering9120757

**Published:** 2022-12-02

**Authors:** Calogero Bugea, Denise Irene Karin Pontoriero, Gaia Rosenberg, Giacomo Mario Gerardo Suardi, Gianmarco Calabria, Eugenio Pedullà, Giusy Rita Maria La Rosa, Francesco Sforza, Antonio Scarano, Roberto Luongo, Giovanni Messina

**Affiliations:** 1Private Practice, Lungomare G. Galilei, 73014 Gallipoli, Italy; 2Department of Prosthodontics, University of Siena, 53100 Siena, Italy; 3Department of Endodontics, University of Genoa, 16126 Genova, Italy; 4Private Practice, Via Ercole Ferrario, 1, 20144 Milano, Italy; 5Private Practice, Residenza Alberata 741 Via Salvo d’Acquisto 4/6, 20079 Basiglio, Italy; 6Private Practice, Via Alessandro Volta 21, 88046 Lamezia Terme, Italy; 7Department of General Surgery and Surgical-Medical Specialties, University of Catania, 95123 Catania, Italy; 8Private Practice, Corso Umberto I 115, 72012 Carovigno, Italy; 9Department of Medical, Oral and Biotechnological Sciences, University of Chieti-Pescara, 66100 Chieti, Italy; 10Private Practice, Via Melo da Bari 229, 70121 Bari, Italy; 11Private Practice, Piazza Primavera 17, 92100 Agrigento, Italy

**Keywords:** root, maxillary premolars, upper premolars, canal, root canal anomalies, anatomical variations, root morphology, coneless, MEA technique

## Abstract

The aim of this case series is to contribute to the better knowledge and management of the complex anatomical configurations of maxillary premolars with four canals. The paper explains the endodontic treatment of five maxillary premolars with four canals, with three buccal and one palatal orifices, in different patients. The cases report several approaches in the treatment of four-canal maxillary premolars including a conservative canal preparation with a hybrid shaping technique, endodontic microsurgery and the application of biomaterials. The use of an operating dental microscope, different operating strategies and the critical evaluation of radiographs are all necessary steps for the correct and safe endodontic management of these teeth.

## 1. Introduction

Understanding the anatomy of the root canal system is a key factor for success in endodontics [1,2,3]. For this reason, clinicians must know all the possible anatomical variations and assess the root and canal configuration prior to, during, and after endodontic treatment. Although the vast majority of maxillary premolars have two root canals, the presence of three distinct root canals has been reported in 1–6% of cases [4,5,6,7,8,9]. Previous case reports have shown several canal and root configurations in three-canal maxillary premolars: three canals in a single root, two canals in the buccal root and one in the palatal root, three separate roots and canals [10,11,12]. Only two studies have described four separate canals in a three-rooted premolar [13,14].

## 2. Materials and Methods

Five patients were found with a rare anatomical variant of a four-canal upper premolar (three first premolars and two s premolar). Each patient agreed to participate and signed a consent form. Assessment of the presence of the number of canals was documented via X-ray examination and clinical photographic evidence.

### 2.1. CASE 1

A 55-year-old male patient diagnosed with pulpitis with no swelling or fistulae. The pre-operative X-ray suggested tooth 14 had an unusual three-rooted anatomy, (Figure 1). After anesthesia and isolation, the whole endodontic procedure was carried out under microscope magnification (PROergo, Carl Zeiss Meditec AG, Munich, Germany) using ultrasonic instrumentation to access cavity refining (no. 3 Start-X, Maillefer Instruments Holding, Ballaigues, Switzerland). A lot of time and dedication was needed in order to identify and negotiate the two buccal canals, due to the closeness of the orifices and their deep location. An intra-operative X-ray was taken to evaluate the root canal system. A three-rooted system was confirmed. Canals were shaped using ProTaper Gold™ (Maillefer Instruments Holding Sàrl, Ballaigues, Switzerland) following the manufacturer’s recommendations. Buccal canals were shaped up to the F1 instrument, while the palatal canal was shaped up to the F2 instrument. Passive ultrasonic irrigation was performed using 5.25% sodium hypochlorite, and a final 10% ethylene-diaminetetracetic acid rinse was carried out at the end. The canals were dried with sterile paper points, and obturated by injection of thermoplastic of gutta-percha using the Obtura III Max (Obtura Spartan Endodontics, Algonquin, IL, USA) in association with an endodontic sealant (Essenseal, Produits Dentaires Sa, Vevey, Switzerland). Upon closer examination, the final X-ray revealed an additional mesial root. After carefully re-examining the pulp chamber, another small orifice, was found very close to the others. This mesial canal was treated in the same session, following the same instrumentation (up to ProTaper Gold, F1), irrigation and obturation protocol used for the other canals. The obturation of the endodontic space was performed by a modification of the injection molding thermoplasticized gutta-percha by Yee [15]. The authors called this kind of obturation Coneless^®^. A final X-ray was taken, and the complete anatomy of this complex endodontic system is clearly shown in Figure 1. The access cavity was filled with a provisional filling material (3M ESPE™ CAVIT™, 3M Italia srl, Pioltello, Italy), and another appointment was scheduled one week later for the definitive restoration. The complete case is illustrated in Figure 1.

### 2.2. CASE 2

A 47-year-old female patient presented to the practice because of swelling in the upper premolar area. Upon intra-oral radiographic examination, a periapical radiolucent area was found at tooth 14, so the patient was scheduled for an endodontic re-treatment. After anesthesia and isolation with a rubber dam, the pulp chamber was accessed, and three canal orifices (one buccal, two palatal) appeared immediately, so as to confirm the suspicion aroused by examination by the pre-operative X-ray. Considering the thickness of the roots, it was useful to prepare the untreated and treated roots of the 14 tooth with the “MEA and inverse taper technique” to keep the dentinal wall removal during the shaping phases under control [16]. The mixed endodontic alloy (MEA) inverse taper^®^ is a hybrid shaping technique comprising the combination of different heat-treated instruments with austenitic files (i.e., Mtwo #10.04 and #15.05, Sweden Martina, Italy) employed in the initial phase of shaping followed by martensitic (i.e., Plex V #20.04 and 25.04, Orodeka, Shandong Province, China) in the second. The first instruments guarantee effective debris removal by means of the more rigid alloy and design; the second ensures improvement in the center ability with respect to the original anatomy. Moreover, the hybrid sequence, constituting the #10.04 and #15.05 files was followed by the #20.04, produced the first taper inversion at 5 mm from the apex while the second phase of #25.04 inverted the taper at 10 mm. It was impossible to completely dry the endodontic system on the first session, so calcium hydroxide was used as an intermediate medication, and the patient was prescribed antibiotic therapy with amoxicillin and clavulanic acid. On the second session, a fourth orifice was identified on the pulp chamber floor in the buccal–mesial root. The fourth canal was itself impossible to dry, so another calcium hydroxide dressing was carried out and the patient was scheduled for a third session, 30 days later, on which more difficulties were met because of the wetting of the apical third of all roots, so an apicoectomy was scheduled. The retrograde preparation was performed with an ultrasonic tip R1D (Piezomed, W&H, Bürmoos, Austria) [16] deep into the canal and obturated with Biodentine as the root-end filling. Seven days after the apicoectomy the patient was recalled for removing the suture and completing the root filling with a single cone and biomaterial (BioRoot™ RCS, Septodont, Saint-Maur-des-Fossés Cedex, France). Several follow-ups were scheduled at different months until the complete healing was obtained and visible by X-ray. The complete case is reported in Figure 2 and Figure 3.

### 2.3. CASE 3

A 40-year-old male patient presented with pain and swelling located in the maxillary projection of tooth 24. An X-ray showed a previous inadequate endodontic treatment and a screw post. Analysis of the pre-operative X-ray suggested a missed canal. After administration with local anesthesia (articaine with 1:100,000 epinephrine) the field was isolated with a rubber dam. All the phases of the procedure, starting from the disassemble of the existing build-up, were performed under the magnification of an operative microscope (PROergo, Carl Zeiss Meditec AG, Germany). The screw post was removed from the palatal canal by means of ultrasound instrumentation (no. 3 Start-X, Maillefer Instruments Holding, Ballaigues, Switzerland).

Removal of the old canal filling began using M-Two Retreatment 25 (Sweden and Martina, Padova, Italy). Working length was determined by an apex locator (Apit Osada, Osada Electric co. Ltd., Tokyo, Japan) and the shaping was finished by Reciproc Blue 25 (Dentsply Sirona, Baden, Switzerland). Irrigation was performed with NaOCl 5.25% (NiClor, Ogna, Bologna, Italy) and activated using the Eddy VDW sonic tip (Dentsply Sirona, Switzerland).

As soon as the chamber and canals were clean, two additional canal orifices in-between were suspected and were easily identified with an endodontic probe DG 16 (Hu-Friedy, Chicago, IL, USA). The initial shaping was performed with M-two Retreatment 15 (Sweden e Martina, Italy) to enlarge the first 5–6 mm. After negotiation and measuring the working length with a manual k-file 10, the M-two 15 0.5 (Sweden e Martina, Padova, Italy) was used to shape up to the working length. All four canals were finished in terms of shaping with an M-two 35 0.4 (Sweden e Martina, Italy). Obturation of the canals was performed using a Bioceramic Cement TotalFill^®^ BC SealerTM (Padova, La Chaux-de-Fonds, FKG Dentaire-Switzerland) with the single cone technique. After the treatment, a post-operative X-ray was taken, and the patient was scheduled for restorative treatment. The complete case is illustrated in Figure 4.

### 2.4. CASE 4

A 54-year-old male patient presented with a spontaneous crown fracture of tooth 25. The intra-oral examination revealed no swelling or sinus tracts, the tooth was vital. The tooth presented an old MOD amalgam restoration and an extended complicated crown fracture involving the palatal half of the crown. The pre-operative X-ray revealed an anatomy of a three-rooted premolar. The whole endodontic procedure was carried out under 4X loupes magnification and dedicated illumination (Carl Zeiss Meditec AG, Germany). After local anesthesia (2% lidocaine with 1:100,000 epinephrine), and rubber dam isolation, a conservative endodontic access cavity was performed using a long-shaft-rounded diamond bur, and dedicated endodontic ultrasonic tips Start X 3 (Maillefer Dentsply, Ballaigues, Switzerland). A careful inspection of the pulp chamber floor revealed an extremely unusual anatomy for a second maxillary premolar, four separate canal orifices were identified. After straight-line access preparation was obtained, root canals were negotiated with pre-curved stainless-steel K-files, sized 0.8 and 10 ISO (Maillefer-Dentsply, Ballaigues, Switzerland), and the WL was established with an apex locator (Root ZX Morita, Tokyo, Japan). A mixed shaping technique was adopted. Pre-flaring and glide path were performed to WL with a NiTi 10.04 and a 15.05 (Sweden e Martina, Padova, Italy) rotary file at 180 rpm and torque 2. All canals were finished with ProTaper Next X2 (Maillefer-Dentsply, Ballaigues, Switzerland). All shaping steps were carried out under 5.25% heated NaOCl irrigation (NiClor, Ogna, Bologna, Italy). After instrumentation, the root canals were irrigated with 17% EDTA solution Tubuliclean (Ogna, Bologna, Italy) for 3 min followed again by several 1-minute rinses with heated 5.25% sodium hypochlorite solution. A carrier-based obturation was performed using dedicated obturators Thermafil for ProTaper NEXT, X2, (Maillefer-Dentsply, Ballaigues, Switzerland) and a zinc oxide-based endodontic sealer. The intra-operative X-ray confirmed all four canals were independent throughout their entire length. A temporary restoration was performed using zinc oxide-based cement placed on the pulp chamber floor covered by a layer of glass ionomer cement (GCem, GC Co Tokyo, Japan). Follow-up after 1 year showed clinical and radiographic signs of healthy conditions. Figure 5 illustrates the complete case.

### 2.5. CASE 5

A 54-year-old male patient was referred for treatment before prosthetic rehabilitation. The pre-operative X-ray revealed the presence of an accessory and an untreated canal of tooth 15, which presented an apical lesion. The whole treatment was conducted under magnification (Leica 525M, Leica Microsystems, Wetzlar, Germany). During disassemble of the cavity, access was refined by ultrasonic instrumentation (Start X no. 2 and 3, Maillefer, Ballaigues, Switzerland) with the aim of locating the missed canal. The treated canals were shaped with a Komet EndoRestart 25/05 (Komet, Besigheim, Germany) and finished with a Protaper Next X2 (Maillefer-Dentsply, Ballaigues, Switzerland). Only the palatal canal was finished with the F3 instrument. The missed canal was then negotiated manually using a k-file 10/02 up to the working length and was then prepared mechanically with the following instrument sequence: M-two 10/04, (Sweden e Martina, Italy), Proglider and Protaper Next X1-X2 (Maillefer-Dentsply, Ballaigues, Switzerland). Warm vertical gutta-percha condensation with a Pulp Canal Sealer EWT (Kerr Dental, Orange, CA, USA) was used for canal obturation. The final X-ray showed unusual anatomy with four independent canals. The complete case is reported in Figure 6.

## 3. Discussion

These cases are rare findings of maxillary premolars with four canals. Endodontic management in these complex anatomical configurations is challenging and depends on individual clinical skills as well as the procedural techniques applied [17].

Accurate radiographic examination using horizontal angle variation provides notable support for the endodontist to distinguish the roots and root canals and formulate a correct diagnosis, also in challenging anatomical configurations, as previously reported [12,13,14]. Furthermore, the detection of the pulp chamber anatomy during coronal access and adequate intracanal dentin removal contribute to the correct clinical location of the root canal orifices [12]. Cone beam-computed tomography (CBCT) is also a diagnostic imaging modality recommended for approaching the complex root canal anatomy [18,19]. Despite CBCT providing high-quality, accurate, three-dimensional (3D) representations of the anatomical dental structures, the presence of metallic restorations (e.g., amalgam restorations, metal posts and/or crowns) or even gutta-percha can determine significant radiographic artefacts which can impact the visualization of the root canal anatomy and pathological conditions, such as root resorption and root fractures [20]. Considering that cone beam-computed tomography implies an additional X-ray dose for the patient, the final decision to perform or not to perform a second-level radiologic exam should be based on the clinical and radiographic interpretation of each case. Within this context, the use of a CBCT exam was not considered pivotal for the management of the clinical cases described.

The microscopy is another essential tool in complex endodontic cases. It favors the localization of anatomical landmarks in the pulp chamber floor and thus the identification of eventual supplementary root canals or root canal aberrations [21]. In some selected cases, the operating microscope can help the clinician in identifying the point where the principal canal bi- or trifurcates and the orientation of the canal orifices [17].

Moreover, in such complex anatomical configurations the use of ultrasonic devices [22], dedicated file for re-treatment [23], and modified preparation techniques [16] could be useful in improving the procedural steps of endodontic treatment.

The mixed endodontic alloy (MEA) inverse taper^®^ is a hybrid shaping technique proposed for the preparation of challenging anatomical configurations, such as double or abrupt curvatures and narrow canals. Inverting the taper during the treatment of a narrow, curved or double canal allows the treatment to be extremely safe. The 0.05% final taper ensures root cleaning until the apex maintaining open the eventual lateral canals. At the same time, the 0.05% taper allows for adequate final shaping for the filling phase. The hybrid sequence comprising the #10.04 and #15.05 austenitic files, followed by the martensitic #20.04 produce the first taper inversion at 5 mm from the apex, while the second phase of #25.04 invert the taper at 10 mm. Thus, the instrument does not engage the dentinal walls at D5 and D10 for #20.04 and #25.04, respectively, avoiding excessive file torsional stress and maintaining as much residual dentin as possible [16].

Finally, the use of straightforward and fast obturation techniques, such as the single cone technique combined with biosealers, which promote hydroxyapatite formation, are particularly useful in challenging root canal configurations [24,25].

These case series provide useful clinical information for performing root canal treatments in complex cases, while creating awareness about anatomical variations of maxillary premolars [26,27,28]. Of note, different clinical outcomes such as orthodontic movement and implant success depend on several factors [29,30], including root canal morphology [31,32]. Hence, a detailed knowledge of root canal morphology is useful in different clinical contexts.

## 4. Conclusions

The above clinical cases describe several approaches in the treatment of four-canal maxillary premolars, including a conservative canal preparation with a hybrid shaping technique, endodontic microsurgery and the application of biomaterials. The use of an operating dental microscope, different operating strategies and the critical evaluation of radiographs are all pivotal steps for the correct and predictable endodontic management of these teeth.

## Figures and Tables

**Figure 1 bioengineering-09-00757-f001:**
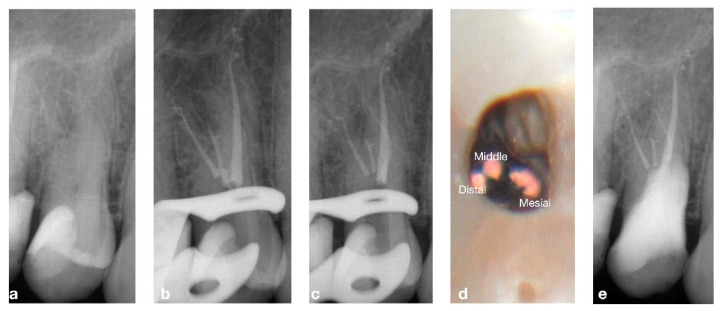
(**a**) Pre-operative X-ray showing the complex anatomy, (**b**) intra-operative X-ray after obturation of three canals, (**c**) intra-operative X-ray after obturation of the 4th canal, (**d**) intra-operative view of the three vestibular canals, (**e**) X-ray of 6-month follow-up.

**Figure 2 bioengineering-09-00757-f002:**
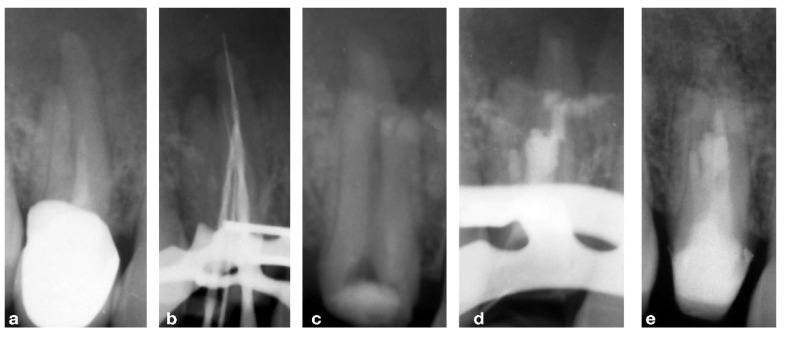
(**a**) Pre-operative X-rays showing the complex anatomy, (**b**) intra-operative X-ray with endodontic instruments to better understand the anatomy, (**c**) post-operative view of the endodontic surgery, (**d**) orthograde filling with gutta-percha, (**e**) 1-year follow-up.

**Figure 3 bioengineering-09-00757-f003:**
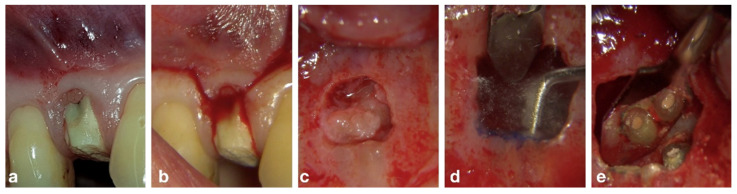
(**a**) Pre-operative view of the premolar, (**b**) flap design, (**c**) intra-operative view of the lesion, (**d**) ultrasonic tip in action, (**e**) intra-operative view of the retrograde obturation.

**Figure 4 bioengineering-09-00757-f004:**
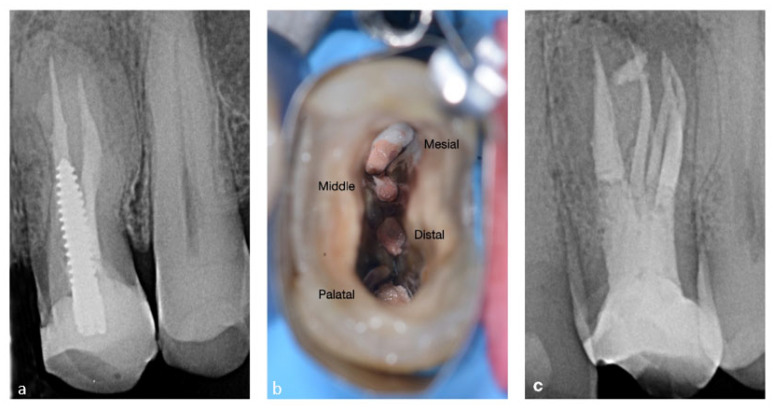
(**a**) Pre-operative view of the premolar, (**b**) post-operative X-ray of the four canals filled with gutta-percha (**c**) 1-year follow-up.

**Figure 5 bioengineering-09-00757-f005:**
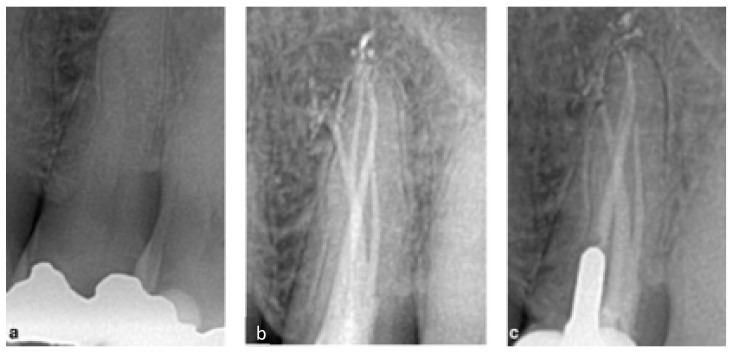
(**a**) Pre-operative view of the premolar, (**b**) post-operative X-ray of the four canals filled with gutta-percha, (**c**) 1-year follow-up.

**Figure 6 bioengineering-09-00757-f006:**
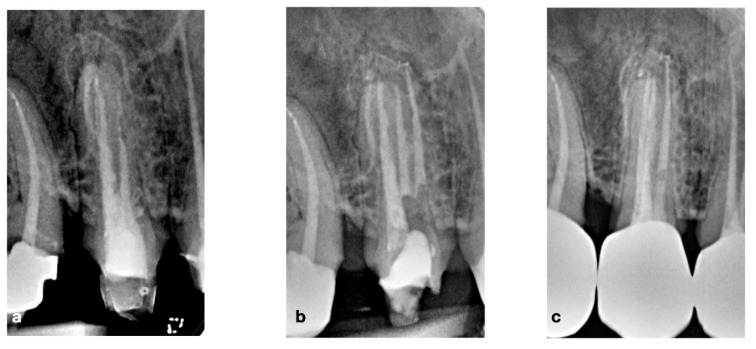
(**a**) Pre-operative view of the premolar, (**b**) post-operative X-ray of the four canals filled with gutta-percha, (**c**) 1-year follow-up.

## Data Availability

Not applicable.

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
