# Peer review of "Maxillary Premolars with Four Canals: Case Series"

_bioengineering, 2022, doi:10.3390/bioengineering9120757_

Round 1

Reviewer 1 Report

In the discussion, the authors state that “it can be helpful for the endodontist,
and the general practitioner, to establish the treatment strategy based on
accurate examination of preoperative and intra-operative X-rays.
I believe that, for complex anatomies, currently, cone beam computed tomography would be the most appropriate exam. The radiographic images presented in this paper do not offer conditions for an adequate analysis.
In case 2, I do not understand how adequate it is to perform the apicectomy and then,
in another session, to fill the canals, or it must be filled before the operation, or a
transoperative filling is performed.

Reviewer 2 Report

The authors present a case series describing five patients presenting maxillary premolars with four canals. Each case management is described, and radiographs and clinical photos are presented to document the cases.

Please check the journal instructions for authors and format the authors' names and affiliations accordingly.

References 10, 11, 12, 17, 18 and 19 are not referenced in the text.

Did you obtain the patients' consent for clinical case publication?

Line 47: I believe you mean figure 1.

I suggest the figures be moved to the appropriate section instead of being presented at the manuscript end. You can check published papers in this journal to help you format your manuscript accordingly.

Images from case 6 are missing.

The presented case series is interesting due to the low frequency of four canal maxillary premolars; however, the manuscript lacks to properly discuss how to diagnose such cases, the advantages and disadvantages of the used techniques and why they are more appropriate for such cases… Additionally, you can discuss how the technological advances/new materials in endodontics are an added value to managing such cases.

I don’t find Bioengineering an appropriate journal for this manuscript. Therefore, I suggest you submit it to a more suitable journal related to the dentistry field.

Reviewer 3 Report

Review on Bugea et al’s Maxillary premolars with 4 canals: case series.

The authors describe a new anatomical variety of maxillary premolars. Basically, the study is well written, but there are a few things that must be corrected.

1.      “Mea and inverse taper technique” is not well known. Despite the study appeared in an Italian journal I tried to access it, but I failed. Please, shortly provide a description of the method used. Though the study is not primarily about the method, but if you mention it: it is good to know what it is all about!

2.      Please, correct this study’s reference, I think there are missing letters from its title! (Instrumentazione…)

3.      The other “problem”: Please, designate the canals on the images from the occlusal view with small arrows and abbreviations. Which is which: P, 2xMB, and DM?

Reviewer 4 Report

The paper is very interesting, however some changes are required before publication

1) please comment if the use of microCT can be useful in diagnosis of such morphology (cite PubMed ID18811596)

2) please comment if such morphology can influence potential orthodontic movements (cite DOI10.1177/1721727X1201000208)3) please comment if , in case of postextractive implants, the prognosis of implant success, due to the socket morphology, can be influenced (please cite DOI10.23805/JO.2018.10.04.04)

Reviewer 5 Report

There are still a lot of issues to work out before this study can be acceptable.

abstract-Cases, treatment procedures, the following results are not well mentioned in the abstract. The abstract should be largely revised to address the significance in this study.

overall comments: This study was designed to present 5 special case treatment reports, however, I was unable to figure out the correct presentation of the data, literature review and discussion.

Round 2

Reviewer 2 Report

The authors present a case series describing five patients presenting maxillary premolars with four canals. Each case management is described, and radiographs and clinical photos are presented to document the cases.

Following previous suggestions, the authors improved the cases description and the discussion section.

Although the reference to a special issue related to Dentistry, I still don’t find Bioengineering an appropriate journal for this manuscript. Therefore, I suggest you submit it to a more suitable journal related to the dentistry field.

Reviewer 5 Report

Since it has been greatly improved after revision, I recommend acceptance for publication.